

# Investigation of lipolytic activity of the red king crab hepatopancreas homogenate by NMR spectroscopy

Maria Timchenko[1], Vladislav Molchanov[2], Maxim Molchanov[1], Alexander Timchenko[3] and Evgeny Sogorin[4]

[1] Laboratory of NMR biosystems, Institute of Theoretical and Experimental Biophysics, Russian Academy of Sciences, Pushchino, Russia
[2] Medical Physics Department, Dubna State University, Branch Protvino, Protvino, Russia
[3] Group of Experimental Research and Engineering Oligomeric Structures, Institute of Protein Research, Russian Academy of Sciences, Pushchino, Russia
[4] Institute for Biological Instrumentation, Federal Research Center ''Pushchino Scientific Center for Biological Research of the Russian Academy of Sciences'', Pushchino, Russia

## ABSTRACT

The digestive gland of craboids (hepatopancreas) is rich in a huge number of various enzymes (collagenases, nucleases, hyaluronidases, proteases), which are well studied at the moment. However, little is known about crustacean lipases. In this work, using $^1$H NMR spectroscopy, it was found that the hepatopancreas homogenate of the red king crab *Paralithodes camtschaticus* demonstrates high lipolytic activity against triacetin in a wide pH range and shows moderate activity against the caprylic/capric triglyceride emulsion. Under the action of the hepatopancreas homogenate, triacylglycerols are converted into 1,2-diacylglycerol, and then into 2-monoacylglycerol and 1-monoacylglycerol. The 1-monoacylglycerol predominates in the reaction products. The use of NMR spectroscopy makes it possible to quickly detect hydrolysis products and evaluate the reaction direction.

## INTRODUCTION

Lipases (triacylglycerol acylhydrolases, E.C. 3.1.1.3) are serine hydrolases; they catalyze the hydrolysis of triacylglycerols to glycerol and free fatty acids (*Sharma, Chisti & Banerjee, 2001*). Lipases are most active upon adsorption onto the oil-water interface. This distinguishes them from esterases, which demonstrate normal Michaelis–Menten kinetics (*Martinelle, Holmquist & Hult, 1995*). The mechanism of interaction of lipase and lipids at the phase interface is still not completely clear and is the subject of intensive research (*Reis et al., 2009*).

Lipases were found in many microorganisms and eukaryotes. To date, most commercial lipases are obtained using microorganisms (*Sztajer, Maliszewska & Wieczorek, 1988*; *Wang et al., 1995*). Lipases are widely used in fats and oils processing, in production of washing powders and grease removers, in food industry and in synthesis of pharmaceuticals, in

Corresponding author
Maria Timchenko,
maria_timchenko@mail.ru

production of paper and cosmetics, and in the waste utilization (*Navvabi et al., 2018*; *Park et al., 2011*; *Zaitsev, Savina & Zaitsev, 2019*; *Masse, Kennedy & Chou, 2001*; *Takamoto et al., 2001*).

Despite the fact that the main function of lipases is the hydrolytic cleavage of triacylglycerol ester bonds, they can also catalyze the reverse reaction (ester synthesis) in a low-water environment. In addition, hydrolysis and esterification can occur simultaneously during transesterification (*Sharma, Chisti & Banerjee, 2001*). Depending on the substrates, lipases can catalyze acidolysis, alcoholysis and transesterification (*Balcão, Paiva & Malcata, 1996*).

The direction of the lipase-catalyzed reaction is determined by the amount of water in the reaction mixture. In the absence or at low water content, only esterification and transesterification reactions occur. When there is an excess of water in the reaction mixture, hydrolysis is the preferred reaction (*Klibanov, 1997*).

The main potential use of lipases is their use as catalysts for a wide range of chemo-, regio- and stereoselective reactions (*Park et al., 2011*; *Zaitsev, Savina & Zaitsev, 2019*; *Berglund & Hult, 2000*). The application of lipases significantly saves energy and prevents the thermal degradation of compounds during the process (*Vulfson, 1994*; *Bornscheuer, 2018*). In addition, most commercial lipase applications do not require high-purity enzymes (*Bayoumi et al., 2007*).

Thus, it can be expected that in the future lipases will have the industrial significance.

One of the application areas of lipases is the production of mono- and diacylglycerols as an alternative to their chemical synthesis (*Bornscheuer et al., 1994*; *Júnior et al., 2018*). Triacylglycerols are sources for the production of monoacylglycerols and diacylglycerols, which are widely used as food emulsifiers (*Kam, Woo & Ong, 2017*; *Jackson & King, 1997*; *Boyle, German & Whelan, 1996*). The mixture of mono- and diacylglycerols is known as food additive E471 (*Plou et al., 1996*).

Commercial species of marine organisms, such as crabs, can be considered as sources of lipases (*Ponomareva et al., 2021*). However, little is known about crustacean lipases. A rich source of highly active enzymes in crabs is the hepatopancreas, which is an organ of the digestive system that performs the functions of the liver and pancreas. Despite the fact that to date, there are data on the investigation of lipolytic activity in various marine organisms (*Smichi et al., 2012*; *Del Monte-Martínez et al., 2019*), only a few research are known on the study of lipolytic activity and lipases from the hepatopancreas of decapod crustaceans, mainly of the infraorder *Brachyura* (*Cherif et al., 2007*; *Cherif & Gargouri, 2009*; *Michiels, Del Valle & Mañanes, 2013*; *Michiels, Del Valle & Mañanes, 2015*). Hepatopancreas lipases for another infraorder of decapod crustaceans *Anomura* have not been practically studied; there are only some mentions of lipolytic activity of hepatopancreas homogenates of pelagic red crab and red king crab without its detailed study (*Del Monte-Martínez et al., 2019*; *Novikov & Mukhin, 2003*). It should be emphasized that hepatopancreas is a waste of crab catching, therefore, the development of methods for processing secondary raw materials in order to obtain new valuable products is an urgent task of rational nature management (*Ponomareva et al., 2021*). Moreover, some hepatopancreas enzymes from red king crab have been found to have unique properties

that are not similar to previously studied enzymes from other marine organisms (*Sliadovskii et al., 2021*). Thus, the investigation of enzymes from crab hepatopancreas can contribute to the study of the dependency of the digestive system on the evolutionary and systematic position of marine species.

In this work, the lipolytic activity of red king crab *Paralithodes camtschaticus* hepatopancreas (HPC) homogenate against short-chain (triacetin) and medium-chain (caprylic/capric triglyceride) triacylglycerols was studied by NMR spectroscopy. The method of NMR spectroscopy facilitates fast estimation of the rate of substrate hydrolysis and analysis of the resulting products under various conditions, which greatly simplifies the study of lipolytic activity.

## MATERIALS & METHODS

### Materials

Triacetin (Polynt UK Ltd., UK) and caprylic/capric triglyceride (IOI Oleo GmbH, Germany) were tested for purity using NMR analysis.

A standard 1M solution of 3-trimethylsilyl-[2,2,3,3-$^2H_4$] sodium propionate (TSP) (Sigma Aldrich, St. Louis, MO, USA) was prepared by dissolving its exact amount in heavy water ($D_2O$, 99.9%; Sigma Aldrich, USA).

Stock solutions of 1 M $Na_2HPO_4$ and $NaH_2PO_4$ (Sigma Aldrich, USA) were used to prepare phosphate buffer solutions with different pH values.

All other chemicals used in this experiment, such as sodium hydroxide, sodium chloride, ammonium sulfate, sodium salt of cholic acid were of analytical grade (Sigma Aldrich, USA).

Pancreatin (OAO THFZ, Russia) purchased in a local pharmacy was used as a positive control for lipolytic activity. The albumin was produced by Amresco (USA).

### Preparation of HPC homogenate

As a hydrolyzing agent, a homogenate of the hepatopancreas of the red king crab *P. camtschaticus* was used, prepared according to the method described earlier (*Ponomareva et al., 2020*). Three samples of HPC homogenate were prepared in a 50 mM phosphate buffer with different pH (pH 5.5, 7.2 and 8.0). The samples were analyzed by Laemmli gel electrophoresis (*Laemmli, 1970*). To assess the lipolytic activity, a freeze-dried HPC sample was also prepared.

### Substrates

For the study, solutions of 330 mM triacetin were prepared in 50 mM phosphate buffer (pH 5.5), 50 mM phosphate buffer (pH 7.2) and 50 mM phosphate buffer (pH 8.0) at 37 °C (*Sigma-Aldrich, 1997*). The resulting solutions were intensively stirred until a homogeneous emulsion was obtained. Triacetin solutions were not adjusted to the desired pH using 1 M NaOH, since this results in the hydrolysis of triacetin (Fig. S1).

To study the kinetics of hydrolysis of triacetin by HPC homogenate, the concentration of triacetin in the reaction mixture was reduced to 70.7 mM (final). At this concentration, the reaction mixture remains optically transparent for two days during NMR analysis.
The value of RG (Receiver Gain) does not exceed the optimal sensitivity of the NMR spectrometer receiver.

Caprylic/capric triglyceride emulsions (18 mM) were prepared in 50 mM phosphate buffer (pH 7.2) containing 150 mM NaCl and 2 mM sodium cholate as a stabilizer. The resulting solution was intensively stirred and sonicated for 10 min under cooling using an ultrasonic disintegrator (operating frequency 22 kHz, UZDN-2T, NPP "Ukrrospribor", Ukraine) for emulsifying the components (*Salentinig et al., 2015*).

## Analysis of lipolytic activity of HPC homogenate
### Sample preparation
To study the efficiency of hydrolysis depending on pH, 750 µl of triacetin (330 mM) was prepared in a 50 mM phosphate buffer at different pH. 250 µl of HPC homogenate was added to the triacetin solution and incubated at 37 °C for 1 h. The reaction was stopped by heating for 5 min at 95 °C. For analysis, 30 µL of $D_2O$ was added to 570 µL of samples. The samples were transferred to an NMR tube and the spectra were recorded. The spectra of triacetin in the absence of HPC homogenate and HPC homogenate at pH 5.5 in the absence of triacetin, which were incubated at 37 °C for 1 h, were also analyzed.

Lipolytic activity was assessed for a lyophilized HPC homogenate according to the Sigma protocol using triacetin as a substrate (*Sigma-Aldrich, 1997*) and by NMR spectroscopy under similar conditions. For the experiments, the same amount of HPC homogenate (in mg) was incubated with the substrate at pH 7.4 and 37 °C for an hour and, subsequently, was analyzed.

The hydrolysis of triacetin with HPC homogenate was studied at pH 7.2. At the first stage, an NMR study was performed with a saturated triacetin solution (*Sigma-Aldrich, 1997*). 25 µl of HPC homogenate solution, 30 µl of 4 mM TSP (in 1 M phosphate buffer, pH 7.2) and 195 µL of 50 mM phosphate buffer (pH 7.2) were added to 750 µL of 330 mM triacetin (in a 50 mM phosphate buffer, pH 7.2). The final volume of the reaction mixture was 1 mL. For NMR analysis, 30 µL of $D_2O$ was added to 570 µL of the mixture. The reaction mixture was incubated at 37 °C directly in the NMR spectrometer. The mixture was incubated for 58 h and NMR spectra were recorded.

As a positive control of lipolytic activity, a solution of pancreatin was used, which was prepared by milling 5 tablets (90 mg each) containing 2,800 units of lipase activity, established by FIP (Federation Internationale Pharmaceutique), and dissolving them in 10 mL of 50 mM phosphate buffer (pH 7.2) followed by dialysis against the same buffer. 25 µl of pancreatin, 30 µl of 4 mM TSP (in 1 M phosphate buffer, pH 7.2) and 195 µL of 50 mM phosphate buffer (pH 7.2) were added to 750 µL of 330 mM triacetin (in a 50 mM phosphate buffer, pH 7.2). The mixture was incubated at 37 °C for 1 h. The reaction was stopped by heating for 5 min at 95 °C. For analysis, 30 µL of $D_2O$ was added to 570 µL of samples. They were transferred to an NMR tube, placed in the NMR spectrometer and the spectra were recorded.

To study the kinetics of hydrolysis of triacetin by HPC homogenate using NMR spectroscopy, the concentration of triacetin was reduced to obtain a solution close to the ideal one. To conduct the reaction, 8 µL of 5.3 M triacetin was added to the solution of

517 µL of distilled water and 60 µL of 4 mM TSP (in 1 M phosphate buffer, pH 7.2), the mixture was intensively stirred and 15 µL of HPC homogenate was added. The sample was gently stirred and the initial time of reaction was fixed. The pH of the reaction mixture at the beginning of the reaction was 6.7. The reaction mixture was incubated at 37 °C directly in the NMR spectrometer. The mixture was incubated for 58 h and NMR spectra were recorded. The pH of the reaction mixture at the end of incubation was 5.9. A similar reaction was conducted with 5-fold increase in the volume of HPC homogenate in the mixture. The resulting reaction mixture was incubated in the NMR spectrometer at 37 °C for 35 h.

The hydrolysis of the caprylic/capric triglyceride emulsion prepared as described above was studied at pH 7.2. 5 µL of 2.4 M caprylic/capric triglyceride was added to the solution of 65 µL of 18.5 mM sodium cholate (final concentration 2 mM), 65 µL of 4 mM TSP (in 1 M phosphate buffer, pH 7.2), 25 µL of 4 M sodium chloride (final concentration 150 mM) and 440 µL distilled water. The mixture was intensively stirred and sonicated, as described above. 50 µL of HPC homogenate was added to the resulting emulsion, gently stirred and the initial time of reaction was fixed. The reaction mixture was incubated at 37 °C directly in the NMR spectrometer. The mixture was incubated for 35 h and NMR spectra were recorded. As control the lipase activity against caprylic/capric triglyceride, a similar reaction was performed with 50 µL of the pancreatin preparation.

### Detection of lipolytic activity

The samples (600 µL) were placed into an NMR tube with a diameter of 5 mm.

The 1D and 2D COSY spectra were recorded on the Bruker 600 AVANCE III NMR spectrometer (The Core Facilities Center of the Institute of Theoretical and Experimental Biophysics of the RAS), operating at a frequency of 600 MHz for protons, using standard pulse sequences from the Bruker pulse sequence library. All measurements were carried out at a temperature of 310 K (37 °C). To suppress the signal from water protons, a pre-saturation method was used by applying a 1D pulse sequence ZGPR. The number of accumulations ranged from 32 to 1,024 scans, the interval between scans was 10 s. The free induction decay (FID) was recorded at 96 k points for 2.272 s. The spectral width was 24 ppm. The duration of the 90°-pulse was 11 µs. After zero-filling of FID to 128 k points the Fourier transform was applied. To study the kinetics of hydrolysis, the spectra were recorded at certain time intervals. The chemical shifts were assigned according to the TSP signal at 0.00 ppm, which acts as an internal reference.

Two-dimensional homonuclear ($^1$H-$^1$H) spin-spin correlation (COSY) spectra were recorded over the range containing signals from 0.15 to 9.15 ppm. During the relaxation delay before the COSY pulses, the water signal was suppressed by pre-saturation. The 2D COSY COSYGPPRQF pulse sequence was used. The relaxation delay between the COSY pulses is 1 s, the data arrays consisted of 2048/512 points.

For signal assignment, one-dimensional NMR spectra and two-dimensional COSY spectra were used. Signal assignment was checked using the AMIX software (Bruker). The spectrum was processed and the integrals were calculated using the TOPSPIN program (Bruker). In the case of hydrolysis of the caprylic/capric triglyceride emulsion, at the end

of the experiment, the sample was removed from the NMR tube, centrifuged at 10,000 $\times g$ for 10 min. The upper hydrophobic layer of the sample was extracted and a deuterated dimethyl sulfoxide solution (DMSO-$d_6$) up to 600 µL was added to it. NMR spectra were recorded using a 1D pulse sequence ZG.

In the case of hydrolysis of triacetin by red king crab hepatopancreas homogenate, the degree of hydrolysis was evaluated (*Nieva-Echevarría et al., 2015*). After the NMR spectra processing the integral values of the components concentrations in the reaction mixture at each time point were calculated based on the integral intensity of proton signals at the second carbon atom of the glycerol part and TSP concentration. The calculation of the components concentrations in the reaction mixture was carried out in the Microsoft Excel spreadsheet software program.

The molar concentration ($N_x$) of 2-monoacetin, 1-monoacetin, 1,2-diacetin and triacetin in the sample can be determined as follows:

$$N_x = C_{st} * n_{st} * I_x / (n_x * I_{st}) \tag{1}$$

where x is the corresponding component of the mixture, $I_x$ is the area under the peak belonging to the signal of the studied proton of the mixture component, $n_x$ is the number of protons providing the signal, $I_{st}$ is the area under the peak corresponding to the proton signal of standard (TSP), $n_{st}$ is the number of TSP protons ($n = 9$), $C_{st}$ is the known concentration of TSP. The area of spectral signal was calculated for the proton at the second carbon atom of the glycerol part of the glycerides, which provides non-overlapping signals in the spectrum.

To obtain the concentration values as close to real as possible, the correction factor between the concentrations of the formed di- and monoacetins from the formed free acetate was calculated. To calculate the real concentration of free acetate formed during the hydrolysis of triacetin, the concentration of the initially present acetate, which is not related to hydrolysis products, was subtracted from the integral concentration of free acetate. The correction factor is the ratio of the real concentration of obtained free acetate to the calculated concentration of acetate obtained from hydrolysis of triacetin, taking into account the integral concentrations of di- and monoacetins.

The concentrations of di- and monoacetins were calculated by multiplying their integral values by the correction factor.

The real concentration of triacetin in the reaction mixture was determined by subtracting the concentrations of the formed di- and monoacetins from the initial triacetin concentration.

The use of this method for calculating the concentrations of components in the reaction mixture is due to the fact that in the process of enzymatic hydrolysis, a solution initially close to ideal loses its homogeneity. The molecules of triacetin, di- and monoacetins in micelles have lower mobility, which affects the decrease in their signals in the NMR spectra. At the same time, a small and charged acetate ion remains in the aqueous phase and partially retains its mobility, which makes it integral concentration closest to the real one. Therefore, all other concentrations were normalized to the acetate concentration.

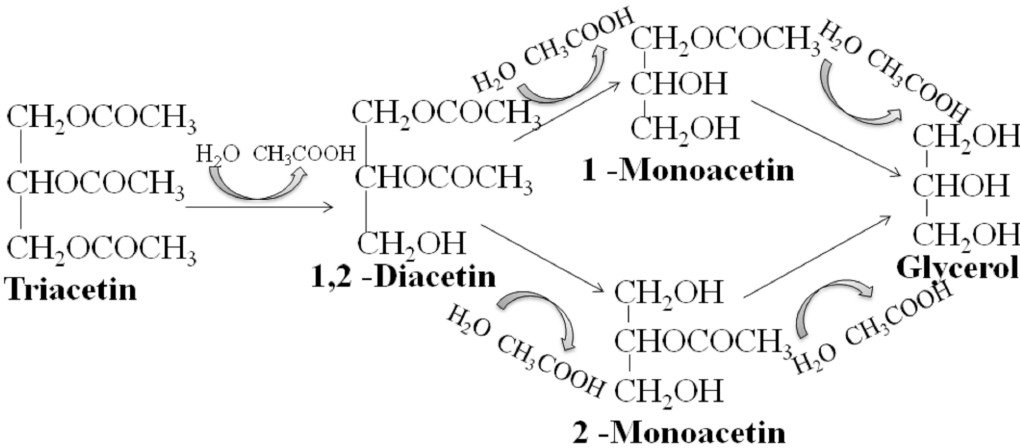

**Figure 1  Scheme of lipase hydrolysis of triacetin.**

The molar percentage of any of the glycerides (X) was determined using the following general equation:

$$N_x(\%) = 100 * (N_x/N_{total}) \tag{2}$$

where $N_x$ is the real molar concentration of the corresponding glyceride, $N_{total}$ is the total amount of glycerides in the reaction mixture.

## RESULTS

We studied the lipolytic activity of HPC homogenate against triacetin by $^1$H NMR spectroscopy method. The general scheme of triacetin hydrolysis is shown in Fig. 1.

As is known, many lipases have a neutral or alkaline pH optimum, in some cases, lipases show maximum activity at pH 9.0 (*Pseudomonas* and *Bacillus* lipases). Acidic lipases are less common, for example, *Pseudomonas fluorescens* SIK W1 lipase has an optimum at pH 4.8. Some *Bacillus sp*. lipases remain active in a wide pH range (pH 3–12) (*Pushkarev et al., 2015*). To analyze the range of lipase activity in the HPC homogenate, triacetin hydrolysis was performed at different pH (5.5, 7.2 and 8.0). Figure 2 shows the $^1$H NMR spectra corresponding to the triacetin hydrolysis by HPC homogenate at different pH for 1 h at 37 °C. The efficiency of hydrolysis was evaluated by a decrease in the proton signal at the second carbon atom in the glycerol part of triacetin approximately at 5.3 ppm and an increase in the signal of the protons of the acetyl group of the formed acetic acid approximately at 1.9–2.0 ppm (Fig. 2, spectra B, C, D). The spectrum of the initial triacetin (before the addition of the homogenate) and the spectrum of HPC homogenate are shown as controls (Fig. 2, spectra A and E, respectively). It was shown that hydrolysis occurs at all the studied pH values. However, in the case of samples with pH 5.5 and 8.0 precipitation was observed. It occurs probably as a result of the HPC proteins precipitation due to the intensive formation of acetic acid and a decrease in pH. For further studies, a pH value of 7.2 was chosen, at which the sample remained stable. It should be noted that the position

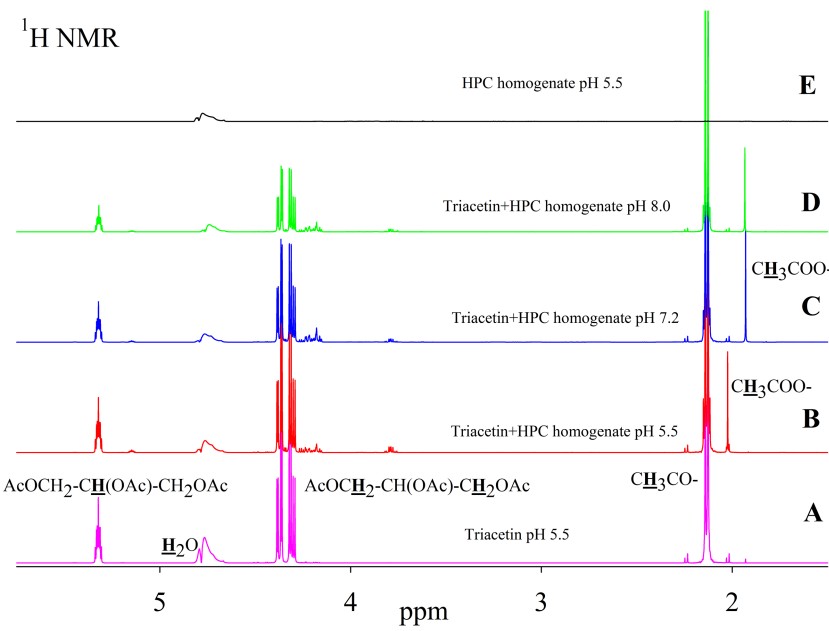

**Figure 2** **¹H NMR spectra of triacetin, HPC homogenate and products of triacetin hydrolysis by HPC homogenate at 37 °C at different pH.** (A) Triacetin (pH 5.5). (B) Triacetin+HPC homogenate (pH 5.5). (C) Triacetin+HPC homogenate (pH 7.2). (D) Triacetin+HPC homogenate (pH 8.0). (E) HPC homogenate (pH 5.5). The protons that give signals are highlighted in bold and underlined.

of the proton signal of the acetyl group of the formed acetic acid strongly depends on the pH, therefore, in the case of an acidic pH, their signal is shifted in the spectrum to a weaker field.

Pancreatin, which includes lipase, was used as a positive control for lipolytic activity (Fig. 3). NMR spectra of its hydrolysis of some triglycerides, for example, tricaprin, were previously described (*Salentinig et al., 2015*). The hydrolysis reaction immediately begins after the addition of pancreatin, which can be detected by the formation of free acetic acid (Fig. 3).

A preliminary assessment of the lipolytic activity for the freeze-dried HPC homogenate by lipase enzymatic assay using triacetin as a substrate (*Sigma-Aldrich, 1997*) showed a value of 150 U/mg, while the value obtained by NMR analysis under similar conditions is 120 U/mg. This correlates with the data obtained by *Novikov & Mukhin (2003)*.

To study the process of triacetin hydrolysis by HPC homogenate at pH 7.2, a saturated triacetin solution was initially prepared (*Sigma-Aldrich, 1997*). After adding the HPC homogenate, the sample was placed into an NMR spectrometer. It took about 10 min to set up the spectrometer and obtain the spectra. During this time, a significant part of the triacetin was already hydrolyzed.

As a result of hydrolysis, triacetin, 1-monoacetin and 2-monoacetin, and 1,2-diacetin were formed (Fig. 4). 1,3-diacetin was not detected during the reaction, which correlates with the literature data obtained for other lipases (*Novikov & Mukhin, 2003*). Free glycerol

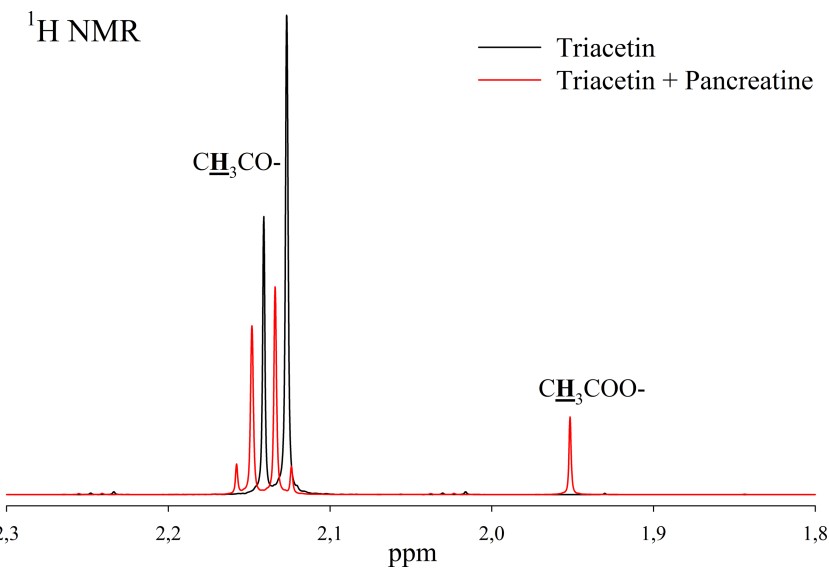

**Figure 3** **$^1$H NMR spectra of products of triacetin hydrolysis by pancreatin at 37 °C and pH 7.2.** Protons giving signals are highlighted in bold and underlined.

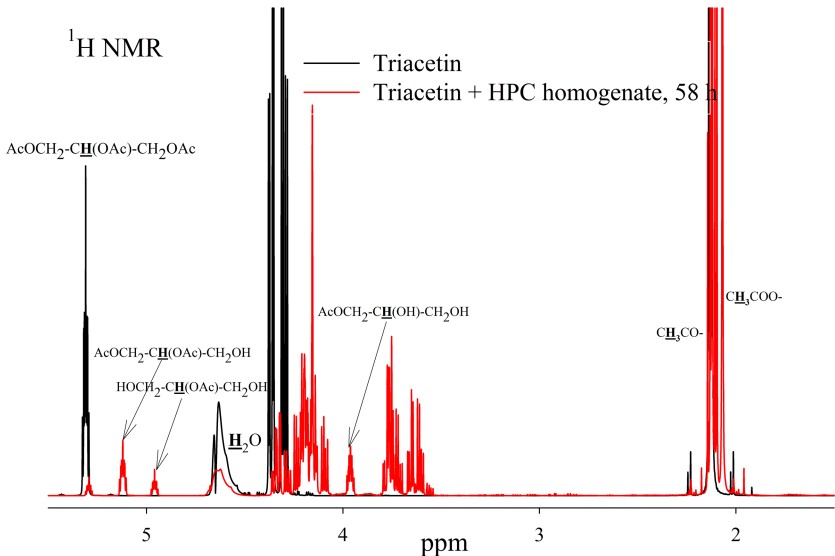

**Figure 4** **$^1$H NMR spectra of products of a saturated triacetin solution hydrolysis by HPC homogenate at 37 °C and pH 7.2 for 58 h.** Ttriacetin:homogenate ratio is 3:1 (by volume). The protons that give signals are highlighted in bold and underlined.

was present in trace amounts. The obtained data were confirmed by the COSY spectra (Fig. S2). A table of chemical shifts is given in the appendix (Table S1).

After the end of the experiment, it was found that a white flake-like precipitate formed in the tube. The precipitate was dissolved in 15 μL of 50 mM phosphate buffer (pH 7.2) with 8 M urea. As a result of the precipitate analysis using Laemmli electrophoresis

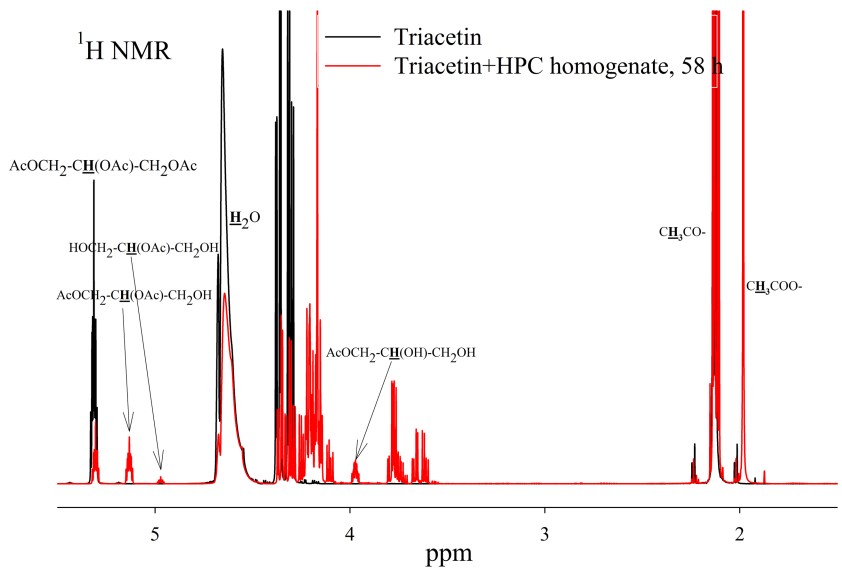

**Figure 5** $^1$**H NMR spectra of hydrolysis products of triacetin solution by HPC homogenate at 37 °C and pH 7.2.** Triacetin:homogenate ratio is 1:2 (by volume). The protons that give signals are highlighted in bold and underlined.

(*Laemmli, 1970*), it was shown that the precipitate is formed by proteins from HPC homogenate (Fig. S3). The pH of the sample after two days of hydrolysis was 3.3. Probably, such a sharp decrease in pH due to the formation of acetic acid led to isoelectric precipitation of proteins from HPC homogenate.

To study the kinetics of triacetin hydrolysis by HPC homogenate, a triacetin concentration of 70.7 mM (below the solubility limit) was used. At this substrate concentration, the reaction mixture remained optically transparent after two days of study, and no precipitate was formed. The volume of HPC homogenate was 10 times less than in the experiment with saturated triacetin (Fig. 5).

To assess the depth of triacetin hydrolysis by HPC homogenate, the percentage of triacetin, 1-monoacetin and 2-monoacetin, and 1,2-diacetin in the mixture was determined based on the concentration of the internal standard (TSP) (Fig. 6). 1,3–diacetin was not detected during hydrolysis, and glycerol was present in trace amounts.

As shown in Fig. 6A, there is an exponential decrease in the triacetin content during hydrolysis by HPC homogenate, which is characteristic of the first-order reaction (Fig. S4). The predominant reaction products are 1,2-diacetin and 1-monoacetin, which is formed from 1,2-diacetin (see Fig. 1). There is also an increase in the content of 2-monoacetin, but its content is much lower compared to 1-monoacetin. With an increase in the concentration of HPC homogenate by 5 times, the rate of the triacetin hydrolysis increases. After 10 h, the concentration of the resulting 1,2-diacetin reaches its maximum value and, further, the decrease of 1,2-diacetin is observed due to the formation of 2-monoacetin and 1-monoacetin. 1-monoacetin is the predominant product of the reaction (Fig. 6B). It should

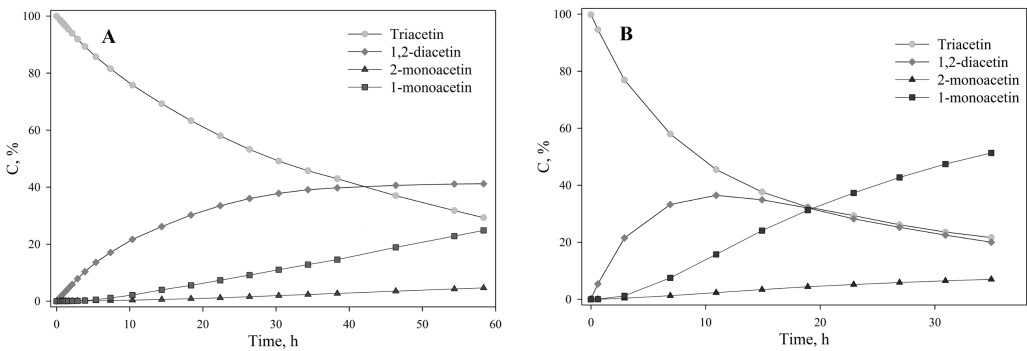

**Figure 6** **The dependence of the percentage of glycerides in the sample on time.** (A) Triacetin:HPC homogenate 1:2 (by volume). (B) Triacetin:HPC homogenate 1:10 (by volume).

be noted that in this case, the dependence of triacetin concentration on time differs from the first-order reaction curve (Fig. S4).

To study the ability of HPC homogenate to hydrolyze medium-chain triacylglycerides, caprylic/capric triglycerides, which are a mixed glycerol ester of caprylic $C_8$ (octanoic) and capric $C_{10}$ (decanoic) acids, were used as a substrate. In the case of medium-chain triglycerides, the preparation of a triglyceride emulsion is the key moment for an effective reaction. Caprylic/capric triglycerides are practically insoluble in water. Thus it was necessary to obtain an emulsion that was prepared using bile acid salts (in our case, sodium cholate) to stabilize the emulsion droplets and modify the oil/water interface for lipase adsorption (*Salentinig et al., 2015*). The mixture was sonicated to obtain a stable emulsion (Fig. S5). In the absence of the enzyme, the resulting emulsion remained stable for several weeks. NMR analysis of the caprylic/capric triglyceride emulsion showed the presence of a triacylglycerol signal at 5.2 ppm as well as the presence of a small fraction of free fatty acids and glycerol admixture, which is stated by the manufacturer of caprylic/capric triglycerides. When HPC homogenate was added to the emulsion, the hydrolysis process began, but the hydrolysis rate was less than in the case of triacetin. Figure 7A shows the NMR spectrum after 35 h of hydrolysis. As seen from the spectrum, diglyceride is formed during hydrolysis. After 35 h of hydrolysis, the initial emulsion was divided into two phases: hydrophobic and hydrophilic (aqueous) (Fig. S5). The reaction proceeded similarly in the case of pancreatin (Fig. S6). To analyze the hydrolysis products, the upper hydrophobic layer of the sample was taken, dissolved in DMSO-$d_6$ and analyzed using NMR spectroscopy. It was shown that the concentration of triacylglycerol in the reaction mixture decreases significantly during 35 h of incubation and the formation of 1,2-diacylglycerol and fatty acids is observed (Fig. 7B).

Thus, the HPC hepatopancreas homogenate shows high lipolytic activity against a short-chain triglyceride - triacetin. The main reaction products are 1,2-diacetin and 1-monoacetin. The formation of predominantly 1-monoacetin distinguishes the HPC homogenate lipase from previously known lipases, for which 2-monoacetin was the predominant hydrolysis product (*Vafaei et al., 2020*; *Lykidis, Mougios & Arzoglou, 1995*).

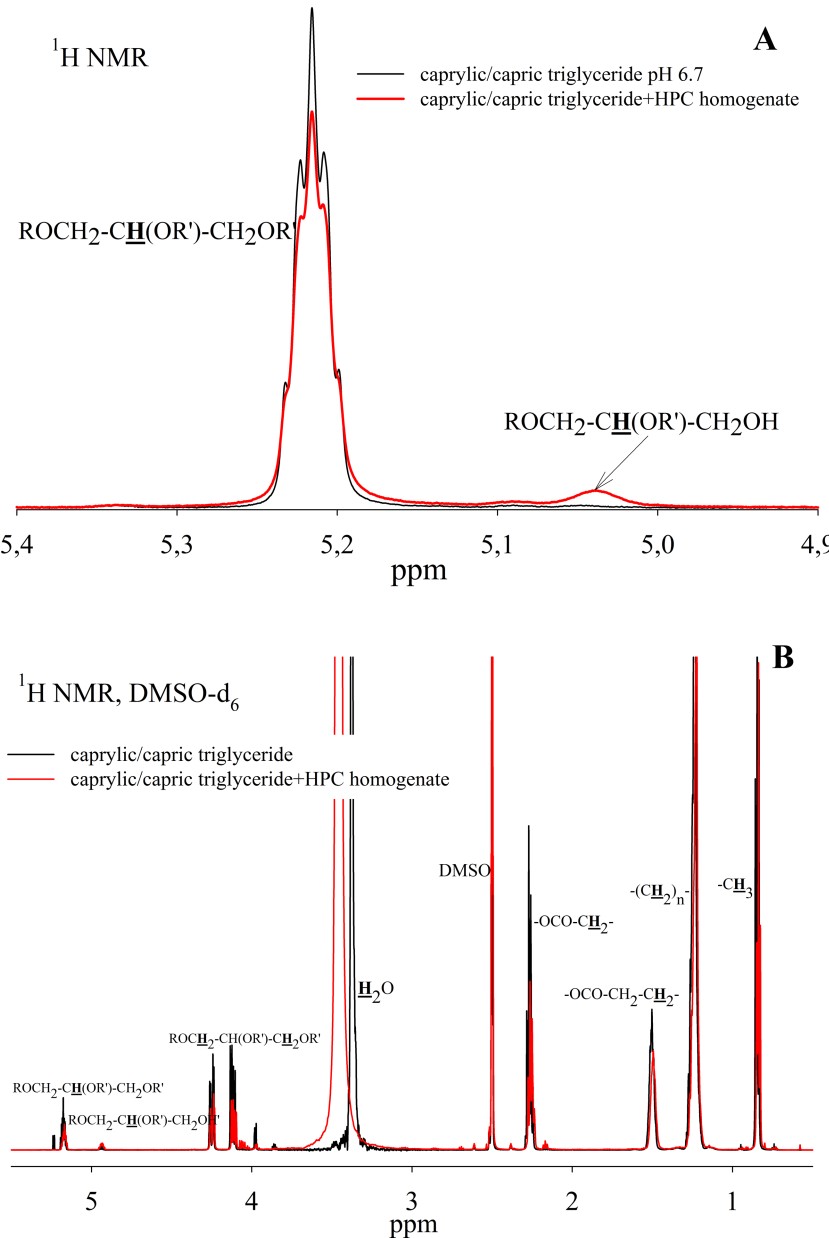

**Figure 7** **$^1$H NMR spectra of a caprylic/capric triglyceride incubated with HPC homogenate at 37 °C and pH 7.2 for 35 h.** (A) The spectrum of the initial triacylglycerol and after hydrolysis. (B) The spectrum of reaction products contained in upper hydrophobic layer which was dissolved in DMSO-d$_6$. The protons that give signals are highlighted in bold and underlined.

Effective cleavage of triacetin by HPC homogenate indicates that the enzyme has free access to the substrate, even in the absence of emulsion stabilizers.

HPC homogenate also shows lipolytic activity against medium-chain triglycerides - caprylic/capric triglyceride. The reaction with HPC homogenate is slower than for triacetin, and the hydrolysis products go into the hydrophobic phase, which makes it

difficult to quantify this process by NMR spectroscopy, but at the qualitative level it is possible to assess the ability of the enzyme to hydrolyze a poorly soluble substrate.

## DISCUSSION

At the moment, not much is known about the study of lipolytic activity and lipases in marine organisms. Lipolytic activity was detected in the gastric juice of the lobster *Homarus americanus* (*Brockerhoff, Hoyle & Hwang, 1970*). In addition, lipase from the hepatopancreas of the green crab *Carcinus mediterraneus* was characterized (*Cherif et al., 2007*; *Cherif & Gargouri, 2009*). There is also mention of the occurrence of lipolytic activity in the crab *P. camtschaticus* (*Novikov & Mukhin, 2003*), but the results of the study are not given. In our work, for the first time, the lipolytic activity of HPC homogenate against certain triglycerides (triacetin and caprylic/capric triglyceride) was investigated.

To monitor the process of lipid hydrolysis, the method of potentiometric titration of released fatty acids using a pH-stat titrator or an autotitrator is used most often (*Brogård et al., 2007*; *Fatouros, Bergenstahl & Mullertz, 2007*; *Helbig et al., 2012*; *Li & McClements, 2010*; *Zhu et al., 2013*). The results obtained by this method are highly dependent on the type of fatty acid in the triglycerides, pH, ionic strength and the concentration of bile salts (*Zhu et al., 2013*; *Sek et al., 2002*; *Thomas et al., 2012*). Various types of chromatography have also been used to quantify lipolysis products, including gas chromatography followed by mass spectrometry (*Helbig et al., 2012*; *Zhu et al., 2013*; *Armand et al., 1999*; *Capolino et al., 2011*; *Hur et al., 2011*; *Meynier et al., 2012*; *Sek, Porter & Charman, 2001*). However, these methods require a large amount of time and a complex sample preparation process. In addition, there is evidence of the non-specificity of the methods and the discrepancy of the results obtained by different authors (*Helbig et al., 2012*; *Zhu et al., 2013*; *Sek et al., 2002*; *Thomas et al., 2012*).

In our work, the NMR spectroscopy method was used to study the lipolytic activity of HPC homogenate. The use of the NMR spectroscopy method made it possible to study the hydrolysis kinetics with high accuracy for stable triacetin solutions and to analyze the resulting hydrolysis products. NMR spectroscopy made it possible to quickly and without preliminary sample preparation to evaluate the activity of the enzyme in relation to the substrate and the ratio between the formed glycerides, which can also be used for other lipases. Previously, the effectiveness of this method for the study of mixtures of tri-, di-, monoglycerides and fatty acids has already been shown (*Salentinig et al., 2015*; *Nieva-Echevarría et al., 2015*; *Gusntone, 1991*; *Ng, 2000*; *Spyros & Dais, 2000*; *Vlahov, 1996*; *Martínez-Yusta & Guillén, 2014*; *Sopelana et al., 2013*; *Gomes et al., 2020*). Prevalently, deuterated chloroform or DMSO (*Nieva-Echevarría et al., 2015*; *Nebel, Mittelbach & Uray, 2008*) were used to study mixtures of glycerides. However as we have shown NMR analysis can be performed in an $H_2O/D_2O$ medium in the case of triacetin.

We have found that HPC homogenate shows high lipolytic activity against triacetin and moderate activity against caprylic/capric triglyceride. It should be emphasized that the mechanism of triglyceride cleavage by HPC homogenate differs from the mechanism of hydrolysis of the most studied lipases: pancreatic lipase (*Vafaei et al., 2020*) and

lipase B from *Candida antarctica* yeast (*Lykidis, Mougios & Arzoglou, 1995*), in which 2-monoacylglycerol is the predominant reaction product.

From an ecological and economical point of view, enzymatic hydrolysis of fats is energetically more profitable than chemical hydrolysis, which requires high reaction temperatures and the presence of catalysts. The chemical hydrolysis makes the process of processing fats complex, toxic and destructive for the equipment (*Boyle, German & Whelan, 1996*; *Plou et al., 1996*). Enzymatic hydrolysis does not require complex equipment and can be used in small and medium-sized manufactures, for example, soap factories. It does not lead to the degradation of fatty acids, as in the case of chemical hydrolysis, and makes it possible to obtain biologically active fatty acids. Lipases have an extremely wide substrate specificity, they are stable and active in organic solvents, do not require cofactors and practically do not give reaction by-products.

Based on the obtained results, it can be assumed that the hepatopancreas of the red king crab can also be considered as a promising source of lipases that can be used to produce various glycerides. It should be noted that a crude hepatopancreas homogenate is sufficient for the reaction. Taking into account the fact that hepatopancreas is a byproduct of processing of red king crab, the catch of which in the Russian Federation in recent decades is 15000–20000 tons per year (*Ponomareva et al., 2021*), it can be assumed that a scalable technology for processing fats can be proposed on the basis of this raw crab material.

## CONCLUSIONS

Thus, for the first time, we characterized the lipolytic activity of the *P. camtschaticus* hepatopancreas homogenate in relation to individual triglycerides. The lipolytic activity of HPC homogenate is preserved in a wide pH range, and the main formed monoglyceride is 1-monoglyceride, which distinguishes HPC lipase from other known lipases that cleave triglycerides with formation of mainly 2-monoglyceride.

## ACKNOWLEDGEMENTS

The authors would like to thank Sergei Lapaev and Azat Abdulatypov for support in preparing this manuscript.

### Funding
The authors received no funding for this work.

### Competing Interests
The authors declare there are no competing interests.

### Author Contributions
- Maria Timchenko conceived and designed the experiments, analyzed the data, prepared figures and/or tables, and approved the final draft.
- Vladislav Molchanov conceived and designed the experiments, performed the experiments, analyzed the data, prepared figures and/or tables, and approved the final draft.
- Maxim Molchanov performed the experiments, analyzed the data, prepared figures and/or tables, and approved the final draft.
- Alexander Timchenko analyzed the data, authored or reviewed drafts of the paper, paper revision, reagents, and approved the final draft.
- Evgeny Sogorin analyzed the data, authored or reviewed drafts of the paper, paper revision, and approved the final draft.

## Data Availability

The raw measurements are available in the Supplementary Files.

## Supplemental Information

Supplemental information for this article can be found online at http://dx.doi.org/10.7717/peerj.12742#supplemental-information.

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
