# Peer review of "Investigation of lipolytic activity of the red king crab hepatopancreas homogenate by NMR spectroscopy"

_PeerJ, doi:10.7717/peerj.12742_

## Round 0.1 · original submission · Minor Revisions

Please provide a comprehensively revised version addressing the editorial comments and a detailed rebuttal letter.

Reviewer 1 ·

Basic reporting

The article clearly describes the procedure to be followed to determine the hydrolysis of ester groups by NMR. However, I consider that it is necessary for you to describe more in the document, the excel sheet developed for the calculations and the methodology followed that they present in the supplementary material.
Include the article: https://doi.org/10.1016/j.rechem.2020.100063
On lines 48 and 49, cite the same reference, correct this.

Experimental design

Correct throughout the document ml by mL

Validity of the findings

no comment

Reviewer 2 ·

Basic reporting

This manuscript by Timchenko et al., discusses the use of NMR spectroscopy to analyze the lipolytic activity of hepatopancreas homogenate from the red king crab Paralithodes camtschaticus. The results are original and well presented and illustrated. However, some comments are made below.

-The authors state that little is known about crustacean lipases. However, although most were done in different Orders that do not include crabs, there are many reports on this subject. This should be clarified in the manuscript.

- Triacetin and caprylic/capric triglyceride are not commonly stored lipids in crustaceans or animals in general. Why did the authors select them for these experiments?

-Because medium-chain triacylglycerides are more abundant than short-chain triacylglycerides. What do the authors think is the biological significance of HPC homogenate showing high lipolytic activity against triacetin and moderate activity against caprylic/capric triglyceride?

-In the last paragraph of the Introduction section the full name of the species used as a study model should be written.

-The manuscript includes sufficient literature references, but many of them are old and should be updated with more current citations. Some examples are:

Reference number 3 can be updated by:
Reis et al., 2009. Lipases at interfaces: A review. Advances in Colloid and Interface Science 147–148, 237–250.

References 6-10 can be updated by:
Zaitsev et al., 2019. Biochemical aspects of lipase immobilization at polysaccharides for biotechnology. Advances in Colloid and Interface Science 272, 102016.

Navvabi et al., 2018. Novel lipases discovery specifically from marine organisms for industrial production and practical applications. Process Biochemistry 70, 61–70.

Kyung-Min et al., 2011. Lipase-catalysed synthesis of erythorbyl laurate in acetonitrile. Food Chemistry 129, 59–63.

-References number 4 and 5 should be relocated to the end of the following sentence: "Commercially significant lipases are usually obtained from micro-organisms that produce a wide range of extracellular lipases [4,5].

-Line 71: other reports on crab lipases should be considered

MichieIs et al., 2013. Effect of environmental salinity and dopamine injections on key digestive enzymes in hepatopáncreas of the euryhaline crab Cyrtograpsus angulatus (Decapoda: Brachyura: Varunidae). Scientia Marina 77(1). doi: 10.3989/scimar.03687.09D.

MichieIs et al., 2015. Lipase activity sensitive to dopamine, glucagon and cyclic amp in the hepatopancreas of the euryhaline burrowing crab Neohelice granulata (Dana, 1851) (Decapoda, Grapsidae).

-Lines 97, 118, 188, 212, 297, 314, 328, 337, 339 replace “red king crab hepatopáncreas” by it abbreviation (HPC).

-Line 145: add a point after .....”15 µl of HPC homogenate was added”

-Line 189: the abbreviation for triacetin (TA) appears for the first time. If this abbreviation is to be used, it should be clarified on line 79 of the Introduction section; otherwise, it should be removed.

-Lines 221-221: "followed by stopping the reaction by heating to 95 °C". This step of the protocol should be clarified in the Materials and Methods section.

-Line 226: replace "enzyme" by "homogenate".

-Lines 238-243 and 282-283 correspond to the Materials and Methods section.

-Line 244 and 261: the abbreviations for 1-monoacetin, 2-monoacetin and 1,2-diacetin have already been defined in lines 189-190. Use them on these lines and line 266.

- When the species is mentioned for the first time, the full name should be given. In the following mentions the name should be abbreviated. See lines 313 and 360.

-The captions of the supplementary figures are missing. In figure S4 it should be clarified which sample corresponds to which lane of the gel.

The format of the listed references needs to be revised and unified. Not all references are presented in the same format.

In captions of figures 2,3,4,5 and 7 the sign ° (°C) should be corrected.

Experimental design

No comment

Validity of the findings

No comment

Additional comments

No comment

·

Basic reporting

The authors described the activity of lipase present at the hepatopancreas of the red king crab Paralithodes camtschaticus. The use of NMR as the assay detection strategy is an increment in comparison to other methodologies previously described. The red king crab hepatopancreas is a waste of the crab food industry and could be a source of lipases with commercial value and industry application. The results also suggest a difference of substrate specificty in comparison to other lipases. Although sequence analysis is not the aim of the manuscript, two literature articles present Paralithodes camtschatichus transcriptome data which could be useful to authors in order to evaluate lipase gene organization and some enzyme properties like molecular mass, pI, terciary structure prediction.


1-Transcriptome profiling and in silico detection of the antimicrobial peptides of red king crab Paralithodes camtschaticus.
Yakovlev IA, Lysøe E, Heldal I, Steen H, Hagen SB, Clarke JL.
Sci Rep. 2020 Jul 29;10(1):12679. doi: 10.1038/s41598-020-69126-4.
2-A crustacean annotated transcriptome (CAT) database.
Nong W, Chai ZYH, Jiang X, Qin J, Ma KY, Chan KM, Chan TF, Chow BKC, Kwan HS, Wong CKC, Qiu JW, Hui JHL, Chu KH.
BMC Genomics. 2020 Jan 9;21(1):32. doi: 10.1186/s12864-019-6433-3.

Experimental design

The experimental design in order to test NMR as an alternative methodology to lipase activity determination was addequate and the supplement material about activity measure was quite informative.
As a suggestion, the analysis of other substrates, including long fatty acid containing substrates and phospholipids. These data would allow the comparison of lipase data with lipases from other Arthropoda species and also could increase the potential industrial application of these lipase.

Validity of the findings

Despite the suggestions on experimental design and on the use of transcriptome data to increment the manuscript, the findings on Paralithodes camtschaticus hepatopancreas lipase and the improvement in lipase measurement activity using NMR contribute to lipase studies and waste potential application.

Additional comments

Although classified as minor revision (due to suggestions previously mentioned) and some English review which could improve text fluidity, the article should be accepted by PeerJ.

·

Basic reporting

The results reported are clear and unambiguous. However, I like to recommend a few corrections in English used.
In line numbers, 103, 126-127, 239, and 280, the authors have tried to add the citation as part of the sentence. That's not the right way. Please complete those sentences without the citation (keep the citation as a citation).
In line numbers, 130-131, it's not clear whether you recorded the NMR spectra or not.
In line number, 146, the sentence in parenthesis "the final concentration of triacetin was 70.7 mM" may be incorrect. Please check and correct it.
In line 159, it's not "To control". it is probably "As control"
In line 170, you stated that "the interval between scans was 10 s" Why such a long relaxation delay?
In line 178, the sentence "The delay time between the COSY pulses was 1 s" may be wrong. You might have meant to say that relaxation delay is 1 s. Change the sentence accordingly.
In line 185, what you meant by "at 10,000 g". Also, it's not "an NMR tube". It is "the NMR tube (in the same line).
In the following line (i.e 186, change "layer of a sample" to "layer of the sample".

Experimental design

no comments

Validity of the findings

no comments

---

## Round 0.2 · Minor Revisions

The manuscript has improved since the last revision. However, it still is mostly qualitative and it does not present quantitative values of the lipase activity. Please estimate the specific activity of the lipase per mg of hepatopancreas, according to the values of activity obtained from NMR. This may look counterintuitive since colorimetric methods for lipase activity have been available for many years. Make a strong case in the conclusions how the NMR methods complement the colorimetric or fluorometric methods to measure lipase activity and provide some values to be compared with the lipolytic activity in shrimp, for example, check https://link.springer.com/article/10.1007%2Fs10126-010-9298-7

---

## Round 0.3 · accepted · Accept

Thanks for addressing the minor revisions requested. Now your manuscript is accepted in PeerJ.